# Hepcidin in Children and Adults with Acute Leukemia or Undergoing Hematopoietic Cell Transplantation: A Systematic Review

**DOI:** 10.3390/cancers14194936

**Published:** 2022-10-08

**Authors:** Artur Słomka, Monika Łęcka, Jan Styczyński

**Affiliations:** 1Department of Pathophysiology, Collegium Medicum Nicolaus Copernicus University Torun, 85-094 Bydgoszcz, Poland; 2Department of Pediatric Hematology and Oncology, Jurasz University Hospital, Collegium Medicum Nicolaus Copernicus University Torun, 85-094 Bydgoszcz, Poland

**Keywords:** hepcidin, iron overload, ferritin

## Abstract

**Simple Summary:**

In this systematic review, we summarized the observational studies on hepcidin in patients treated for acute leukemia or undergoing hematopoietic cell transplantation (AL/HCT). Thanks to the rigorous methodology used, we were able to trace the available literature conscientiously and draw the following conclusions: (1) in both children and adults with AL and qualified for HCT, hepcidin levels are high regardless of the phase of the disease or iron resources;, (2) AL therapy, and HCT in particular, may affect hepcidin levels, but the data, especially for children, are fragmented; (3) pre-HCT hepcidin levels may help predict post-HCT outcomes; (4) there is a need to standardize the determination of hepcidin levels in the clinical setting. We find a very large discrepancy in the reported mean and median hepcidin levels, both in healthy subjects and in AL. This significantly hinders the interpretation and comparison of the results.

**Abstract:**

Objectives: The association between hepcidin and acute leukemia (AL) or hematopoietic cell transplantation (HCT) in children and adults remains obscure. We aimed to assess this potential relationship through a systematic review of observational studies. Methods: An electronic search of three databases, including PubMed, Scopus, and Web of Science Core Collection, was performed up to 31 March 2022. Two independent reviewers assessed the search results according to predetermined inclusion and exclusion criteria, following PRISMA guidelines. Results: Of the 3607 titles identified, 13 studies published between 2008 and 2021 met the inclusion criteria. Most studies included a moderate number of participants and controls and used enzyme-linked immunosorbent assay (ELISA) to determine serum hepcidin levels. The principal findings: (1) serum hepcidin levels in patients with AL or undergoing HCT are increased compared to controls, regardless of the patient’s age and the phase of disease treatment; (2) AL therapy and HCT significantly influence serum hepcidin levels; (3) serum hepcidin may predict a worse outcome in patients with AL and post-HCT. Conclusions: This systematic review provides an overview of observational studies that deal with the association of hepcidin with AL and HCT. Although disturbances in iron metabolism are common in AL and HCT, and hepcidin seems to play a cardinal role in their modulation, more extensive research is needed.

## 1. Introduction

Acute leukemias (AL) are belligerently progressive neoplasms of the bone marrow, characterized by clonal expansion of immature and highly undifferentiated hematopoietic cells [1,2]. In conformity with the 2020 Global Cancer Statistics study, leukemias were diagnosed worldwide in over 474,000 patients with over 311,000 mortalities [3]. Despite substantial progress in AL patient management, these diseases remain a grievous clinical concern in pediatric and adult groups [4,5].

It is becoming increasingly clear that iron overload is an exceptionally influential component of the pathophysiology of AL, and, as recent studies show, this over-abundance is a fundamental factor that may negatively affect the outcome of patients [6,7]. Although iron overload is observed in some patients with AL at diagnosis [8,9], the leading causes of this phenomenon are frequent blood transfusions [10,11] during chemotherapy treatment [12,13,14,15]. Notably, many investigations revealed a close relationship between iron overload—primarily defined as hyperferritinemia—and poor prognosis in patients undergoing hematopoietic cell transplantation (HCT) [8,16,17]. This was also confirmed in a meta-analysis of 25 studies with 4545 patients undergoing HCT, which demonstrated that pre-transplantation hyperferritinemia has a negative prognostic role and is associated with decreased overall survival and progression-free survival, as well as a higher incidence of non-relapse mortality and bloodstream infections [18].

Due to the extreme toxicity of iron in the human body, an immeasurably precise mechanism operates to control its levels inside and outside of cells. Many proteins are involved in this machinery, but hepcidin is of crucial importance. Produced by the hepatocytes, this small 25-amino acids protein inhibits iron absorption and releases from tissue resources by degrading ferroportin, the sole known cellular iron exporter [19]. Considering the paramount role of hepcidin in iron metabolism and the disturbances noted in AL and after HCT, it is understandable that researchers were looking for the role of hepcidin in the course of diseases.

Studies published to date, also by our team, suggest that hepcidin levels are high in both children and adults with AL [20,21]. Most studies, however, focus on assessing hepcidin levels in HCT patients as a predictor of adverse events such as infections, acute graft-versus-host disease (aGVHD), and poor overall survival [22,23,24]. That notwithstanding, the disorganized data on the relationship between hepcidin, AL, and HCT may cause an earnest misinterpretation of the role of hepcidin in the pathophysiology and prognosis of the disease process or the post-transplant patient response.

To clarify this issue, we performed a systematic review of studies reporting the association between hepcidin levels, AL, and HCT. As far as we are aware, our study is the first that exclusively focuses on the assessment of hepcidin in these patient groups.

## 2. Materials and Methods

### 2.1. Search Methodology

We conducted a systematic review following the Preferred Reporting Items for Systematic Reviews and Meta-Analyses (PRISMA) statement (Appendix A) [25] and prospectively registered the review on PROSPERO (identifier: CRD42022323952). 

We comprehensively searched the PubMed, Scopus, and Web of Science Core Collection electronic databases to identify studies published until 31 March 2022 (date of the last search), with detailed search terms for: “hepcidin”, “leukemia”, and “hematopoietic stem cell transplantation”. The full PubMed search strategy is shown in Appendix A and was appropriately translated for the other two databases. There were no restrictions on language or publication type. We also hand-searched the bibliographies of all the included studies and relevant review articles to identify any remaining studies. Duplicate studies were manually deleted using Zotero, version 6.0.4 (Corporation for Digital Scholarship, Vienna, VA, USA).

### 2.2. Study Selection

Two reviewers (A.S. and M.Ł.) made the initial selection of the studies based on titles and abstracts. Next, we obtained the full texts of the studies that seemed to fulfill the inclusion criteria for evaluation. Any discrepancies were resolved by consensus, referring back to the studies, in consultation with a third reviewer (J.S.).

### 2.3. Inclusion Criteria

Studies were deemed eligible if they met the following criteria: (1) were observational (case-control, cohort, or cross-sectional studies) [26] and (2) included original data relevant to measuring hepcidin levels in pediatric and adult AL patients or those undergoing HCT. Stages of disease or treatment modalities were not the criteria for excluding the study from the systematic review. We did not set a minimum number of patients as a criterion for a study’s inclusion; however, case reports were not included in the systematic review.

### 2.4. Exclusion Criteria

Studies were excluded if they included patients with diseases other than AL—unless patients with AL constituted only a part of the study group. When AL patients comprised only a portion of the study group, we included such studies in the systematic review. For example, if the study group included patients with AL and myelodysplastic syndromes (MDS), the study was included in the current systematic review. Other exclusion criteria were insufficient data on hepcidin (e.g., lack of numerical values of hepcidin levels) and studies published in non-English languages. Clinical trials, reviews, case reports, editorials, comments, position articles, guidelines, chapters of books, conference proceedings, and nonhuman studies were also excluded.

### 2.5. Data Extraction

Relevant data from the included studies were extracted by two reviewers (M.Ł. and J.S.). From each eligible study, data were captured on the following: general study information (first author, year of publication, study design, and study location), participant characteristics (sample size, age, sex, diagnosis, and therapeutic modalities), details relating to the assessment of hepcidin (type of biospecimen, measurement time with corresponding detection method, and hepcidin levels), and, lastly, each study’s main findings. All data were extracted from the published studies; we did not contact the corresponding authors to collect further information.

### 2.6. Quality Assessment

We assessed the methodological quality of the included studies by using the Newcastle–Ottawa (NOS) scale for case-control and cohort studies [27], adapted for cross-sectional studies [28]. The NOS score is recommended for assessing the quality of nonrandomized studies [29]. As originally developed, NOS consists of eight items in three domains: selection (total score 4), comparability (total score 2), and exposure for case-control studies or outcome for cohort studies (total score 3). The highest total score is nine points. The NOS, as adapted for cross-sectional studies, consists of seven items in three domains: selection (total score 5), comparability (total score 2), and outcome (total score 3), with the highest total score being ten points. A total score of 3 or less was considered low quality, 4 to 6 was considered moderate quality, and 7 to 9 (10 for cross-sectional studies) was deemed high quality [30]. Any discrepancies in the quality assessment were discussed and resolved by the two reviewers (A.S. and M.Ł.).

## 3. Results

### 3.1. Literature Search

Our systematic search identified 3607 unique citations. Of these, 3439 (95%) were found in Scopus, 106 (3%) were found in Web of Science, and 62 (2%) were found in PubMed. No additional studies were identified through our hand search of references from published studies, relevant reviews, and previous meta-analyses.

After adjusting for duplicates, the searches provided a total of 2949 citations, of which 2802 were excluded based on abstract and title. One hundred forty-seven articles underwent full-text review, and one hundred thirty-four were excluded. The remaining thirteen met all inclusion criteria [20,21,22,24,31,32,33,34,35,36,37,38,39]. There was no disagreement about any of the studies selected for final inclusion in the systematic review. Most studies were excluded because they were irrelevant to the current study subject (*n* = 45). A full list of excluded studies and reasons for exclusion is available in Appendix A. Due to the study design’s high heterogeneity, we did not use formal meta-analysis techniques. The flow of study selection is reported in Figure 1.

### 3.2. Characteristics of the Included Studies

The characteristics of the studies included in the review are given in Table 1 for case-control studies and in Table 2 for cohort and cross-sectional studies. The studies were published between 2008 and 2021 with 92% (*n* = 12) of the studies published on or before 2020 [20,22,24,31,32,33,34,35,36,37,38,39]. Out of the thirteen studies [20,21,22,24,31,32,33,34,35,36,37,38,39], three were conducted in Japan [22,24,31], two in Germany [32,38], two in China [20,34], and one each in the United States [33], Brazil [35], Egypt [36], Turkey [37], Indonesia [39], and Poland [21]. Eleven studies were single-center [20,22,24,31,32,33,34,35,36,37,39] while two were multicenter [21,38]. We included seven case-control studies [20,21,31,32,34,36,37], four cohort studies [22,24,35,38] and two cross-sectional studies [33,39]. Eleven studies were fully published journal articles [20,21,22,24,32,33,34,36,37,38,39]; one study was published as a brief report [31], and one as a letter to the editor [35].

All studies combined, there were a total of 781 patients and 154 controls. Most patients were male [20,21,24,32,34,36,37,38,39]. In four studies [37,38,40,41], participants were children and adolescents, with the mean or median age ranging from 5 [36] to 10 [37] years old; all participants in the other studies [20,22,24,31,32,33,34,35,38] were adults, and the mean or median age ranged from 29 [20] to 62 [38] years old. Participants represent heterogeneous populations; however, most studies included patients with acute myeloid leukemia (AML) [20,22,31,32,33,38]. None of the included studies describe the selection of controls; hence, it is unknown whether the participants were population- or hospital-based controls. Follow-up varied between studies, ranging from 100 days [22] to 46.8 months [24]. One study reported follow-up as documented engraftment [35]. The patient populations were HCT in nine studies [21,22,24,31,32,33,34,35,38]. Five studies were limited to allogeneic HCT [22,24,32,33,38], one study was limited to autologous HCT [35], and the remaining three studies included both autologous and allogeneic HCT [21,31,34]. Only one study included pediatric HCT patients [21].

### 3.3. Quality Assessment of Studies

We assessed the quality of the thirteen eligible studies according to the NOS (Appendix A). We judged four studies [20,21,32,36] to be of high methodologic quality and the remaining nine studies [22,24,31,33,34,35,37,38,39] to be of moderate quality. No studies were of low quality. The median NOS score was 6 (range: 5 to 7).

### 3.4. Hepcidin Assays

All included studies specified the time at which biological samples were obtained. Eight studies have documented pre-transplant hepcidin levels [22,24,31,32,33,34,35,38], six post-transplant hepcidin levels [21,31,32,34,35,38], four before treatment [20,21,36,37], two during treatment [36,39], three at remission [20,36,37], and one after treatment [21]. Five studies had three time points of assessment [20,34,35,38,39]. Data on the collection, preparation, or storage of biological samples were described in five studies [21,22,24,33,37]. For the testing of biological materials, twelve of the studies evaluated hepcidin levels in the blood (serum or plasma) [20,21,22,24,31,32,34,35,36,37,38,39] and the other one in blood and urine [33]. Different hepcidin assays were applied in the selected studies. Nine studies used enzyme-linked immunosorbent assay (ELISA) kits from different suppliers [20,21,32,34,35,36,37,38,39]. However, the most commonly employed kit in the four studies was DRG Instruments GmbH (Marburg, Germany) [34,35,37,38]. The mass spectrometry (MS) methods were also used to measure hepcidin levels in four studies [22,24,31,33]. Most of the selected studies did not provide sufficient data regarding the methods applied for hepcidin measurement. Only one study reported assay range, detection limit, and intra- and inter-assay coefficient of variation (CV) for hepcidin assay [21]. Two studies reported a normal range for serum hepcidin, according to manufacturers [32,38]. Blinding of laboratory personnel to the clinical characteristics and patients’ outcomes was reported in one study [21]. For the standardization of the results and straightforward comparisons between studies in this systematic review, the hepcidin levels are presented in nanograms per milliliter (ng/mL) for each study included.

Table 3 compares blood hepcidin levels with a commonly used marker illustrating iron metabolism, i.e., serum ferritin levels. As expected, ferritin levels were very high in the patients in the included studies, which is also related to the number of packed red blood cells (PRBCs) transfused (Table 3).

### 3.5. Hepcidin Levels in Childhood Leukemia

Three case-control studies [21,36,37] and one cross-sectional study [39] evaluated hepcidin levels in childhood leukemia. The studies included ranged in size from 40 [36] to 67 patients [21] and from 17 [37] to 20 controls [36], for a total number of 213 patients and 55 controls. Most of the included patients were diagnosed with acute lymphoblastic leukemia (ALL). The main message from case-control studies is that hepcidin levels are significantly higher in children with AL than in controls, regardless of the stage of the disease [21,36,37]. Hepcidin levels vary between phases of the disease [21,36,37,39], and they appear to decrease for cases of childhood leukemia in remission when compared to the levels at the time of diagnosis [36,37]; however, a significant difference was found only in one study [36]. Two studies demonstrated that hepcidin levels are lower during maintenance therapy [36,39]. Particular attention should also be paid to the wide range of serum hepcidin levels in both AL and controls (Table 1), e.g., 58.45 ng/mL [37] to 387.6 ng/mL [36] at the diagnosis of the disease. In the controls, the differences are even more noticeable (ten times lower levels in one study [37] compared to another [36]). This is a supplemental factor, due to which we did not perform a meta-analysis.

### 3.6. Hepcidin Levels in Adult Leukemia

Four case-control studies [20,31,32,34], four cohort studies [22,24,35,38], and one cross-sectional study [33] were used to evaluate hepcidin levels in adulthood leukemia. The studies included ranged in size from 31 [31] to 166 patients [24] and from 17 [31] to 50 controls [34], for a total number of 568 patients and 99 controls. In the case of studies in adult patients, we found significant heterogeneity in the populations; however, AML seems to be the most frequently diagnosed.

The main message from case-control studies performed in adults with AL is similar to the conclusion involving childhood studies. Hepcidin levels are significantly higher in adults with AL than in the controls, regardless of the stage of the disease or of patients’ iron storage [20,31,32,34]. A single study found a decrease in serum hepcidin levels in patients in remission compared to pre-treatment levels [20]. As the remaining studies [22,24,31,32,33,35,38] largely concerned patients undergoing HCT, we described their results in the next subsection of our systematic review. As in the pediatric population, studies in adult patients and adult controls showed a wide spread of the mean or median of hepcidin levels (Table 1 and Table 2).

### 3.7. Hepcidin Levels in HCT Patients

HCT was performed in adult patients in eight studies (*n* = 498) [22,24,31,32,33,34,35,38]. In two studies, a significant increase in serum hepcidin levels was observed one week after HCT compared to pretransplant levels, with normalization of the levels one month after transplantation [31,34]. One study found no differences in the serum levels of hepcidin 10 days before HCT compared to the third month afterward [32]. In turn, the lowest level of serum hepcidin was found before the start of conditioning rather than before stem cell (SC) infusion or on engraftment [35]. High serum hepcidin levels before transplantation are also associated with a higher risk of bacterial infections [22], invasive fungal disease [34], and lower overall survival [24] after HCT. The remaining studies found a positive relationship between pre-transplant serum hepcidin levels and other markers of iron metabolism [33,38]. 

Only one study investigated hepcidin levels in children undergoing HCT (*n* = 21): it found that one month after transplantation, serum hepcidin levels were significantly higher compared to the levels at diagnosis or after the end of intensive chemotherapeutic treatment [21].

## 4. Discussion

Iron overload is a common secondary complication in patients treated for AL or undergoing HCT and is caused by frequent packed red blood cell concentrate (PRBC) transfusions. Each milliliter of transfused PRBC contains 0.8 mg of iron [40]; thus, repeated transfusions ponderously contribute to iron accumulation. No other diagnoses in oncology bearing this complication in such an aggrandized grade. Iron overload is the long-term sequelae of blood component therapy, and intensive treatment damages cells, causing clinically relevant homeostasis imbalance. The pathophysiological processes following one another include: PRBC repeated transfusions; iron delivery; ferritin production and storage; iron overload; imbalance in iron regulation; cellular, tissue, and organ toxicity; and finally, organ failure. 

There are four significant findings of our study. First, hepcidin increases during intensive chemotherapy of AL, then partially decreases during maintenance therapy or after its completion. Second, in patients undergoing allogeneic HCT, hepcidin increases during the pre-engraftment period and might partially decrease after engraftment. Third, these profiles of serum hepcidin levels seem to be similar in children and adults. Fourth, hepcidin levels correlate with ferritin levels and iron overload status in patients treated for leukemia or undergoing allo-HCT.

From a pathophysiological point of view, hepcidin production in hepatocytes is stimulated by various factors, including iron and inflammatory status, expressed substantially by the upregulation of interleukin-6 (IL-6). The serum levels of hepcidin correlate with the serum levels of ferritin, and both proteins are upregulated by systemic iron overload. In this context, an increase in serum hepcidin levels reflects a regulation mechanism secondary to iron overload. Therefore, it is unsurprising that the ongoing disease process and applied treatment induce inflammation, leading to increased hepcidin synthesis. As such, our observation of high serum hepcidin levels is widely reported in children and adults. New data emerging from our analysis shows that, regardless of the phase of the disease (and thus also of treatment), hepcidin levels remain high in patients compared to controls.

Increased serum hepcidin levels are also the result of the compensation of iron overload. Nevertheless, mechanisms of homeostasis, and return of effective myelopoiesis, including erythropoiesis, cannot cause full utilization of iron excess. In this context, hepcidin in patients treated for AL or undergoing HCT is an ineffective marker of iron overload and metabolism. 

No studies directly compare hepcidin levels in children and adults with AL. The profile of hepcidin levels during and after intensive anti-leukemic chemotherapy seems similar and age-independent. However, it is possible that mechanisms of homeostasis and organ abilities to compensate for organ toxicities are better in children than in adults; thus, some improvement, expressed as a decrease in iron overload, ferritin or hepcidin levels, can be expected in the pediatric population [41,42,43]. 

Hepcidin levels peaked after the conditioning and pre-engraftment phase in transplanted patients, then decreased during engraftment [44]; however, not in each study [35]. This process clinically correlates with the need for frequent blood transfusions before engraftment, which is not the case afterward due to more efficient myelopoiesis and partial iron utilization. 

HCT is a perfect model showing the profile of changes in serum hepcidin levels related to iron overload, followed by effective erythropoiesis, presented in three phases. The first phase, on patients’ referral to HCT, reflects the status of patients heavily transfused with PRBC, often in chronic inflammatory status caused by chemotherapy-induced mucositis and possible organ toxicity. Thus, pre-transplant serum hepcidin levels are usually high, at least doubled in most studies, compared to leukemic patients on diagnosis, both in children and adults [21,31,34]. In the second phase, during the conditioning and pre-engraftment phase, the serum hepcidin levels usually increase [34,35], as the intensity of PRBC transfusion is even higher than in non-transplant patients. High hepcidin levels cause a delay in platelet engraftment after HCT [24]. Finally, the third phase starts from the day of engraftment, followed by effective erythropoiesis, resulting in the utilization of iron and decreased ferritin and hepcidin levels. Obviously, it is not possible to utilize all excesses of iron; hence, the status of iron overload persists, causing cellular, tissue, and organ damage impairing their function and leading to worse overall survival [14,24]. In the case of non-transplant leukemic patients, this becomes a two-phasic model. The first phase, during intensive chemotherapy, is characterized by frequent PRBC transfusions and increasing iron overload, followed by increased ferritin and hepcidin levels [21,36]. In the second phase, during maintenance therapy or after cessation of treatment, when effective erythropoiesis is present, the content of iron and levels of ferritin and hepcidin is lowered [36]. 

The strength of the study is the first systematic review on hepcidin in AL/HCT patients with a new insight into the assessment of serum hepcidin levels in pediatric and adult patients, showing variability in serum profile before, during, and after AL/HCT treatment.

This study has several limitations due to heterogeneity of the studies, relatively low number of patients included in basic studies, lack of studies comparing other novel parameters of iron metabolism, lack of studies comparing children and adults, and lack of long-term analyses of survival outcomes. 

## 5. Conclusions

To the best of our knowledge, our study is the first to summarize the observational studies on hepcidin in AL and HCT. Thanks to the rigorous methodology used, we were able to trace the available literature conscientiously and draw the following conclusions: (1) in both children and adults with AL and qualified for HCT, hepcidin levels are high regardless of the phase of the disease or iron resources; (2) AL therapy, and HCT in particular, may affect hepcidin levels, but the data, especially for children, are fragmented; (3) pre-HCT hepcidin levels may help predict post-HCT outcomes; (4) there is a need to standardize the determination of hepcidin levels in the clinical setting. We find a very large discrepancy in the reported mean and median hepcidin levels, both in healthy subjects and in AL. This significantly hinders the interpretation and comparison of the results.

In conclusion, we can show that the profile of hepcidin levels in patients treated for AL/HCT is presumably similar in children and adults. Hepcidin levels increase relatively quickly with RBC transfusions during intensive chemotherapy for AL or between the start of conditioning and engraftment after HCT. However, in both settings, it tends to decrease: during maintenance therapy for AL or in the post-engraftment phase of HCT. Nevertheless, the homeostasis mechanisms are not efficient enough, and increased hepcidin levels might be a risk factor for overall survival.

## Figures and Tables

**Figure 1 cancers-14-04936-f001:**
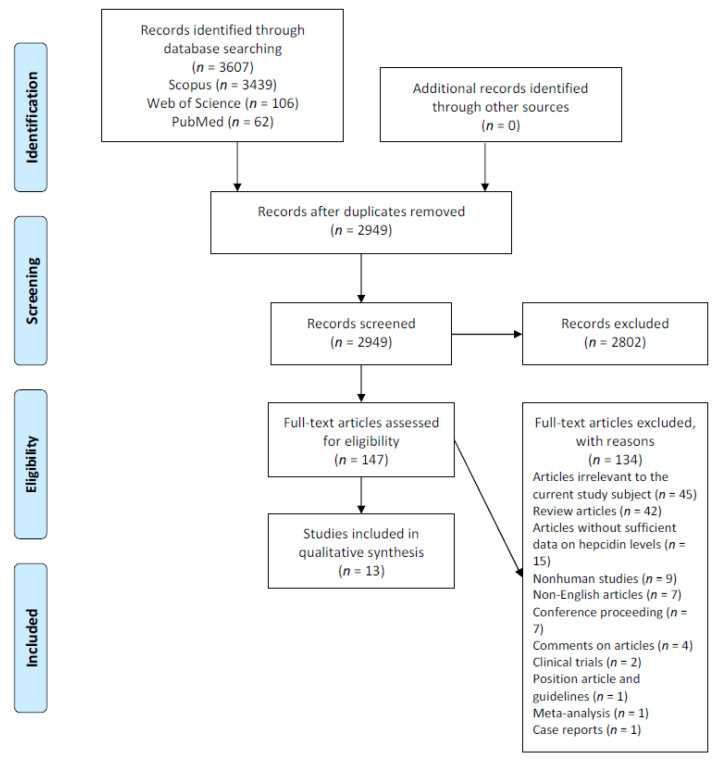
Preferred Reporting Items for Systematic Reviews and Meta-Analyses (PRISMA) flow diagram.

**Table 1 cancers-14-04936-t001:** Characteristics of the seven case-control studies included in a systematic review.

First Author, Year	Region of Origin	Number of Patients	Mean or Median Age of Patients [Years]	Clinical Diagnosis	HCT	Hepcidin Assay	Type of Biological Material	Mean or Median Hepcidin Levels in Patients	Number of Controls	The Mean or Median Age of Controls [Years]	Mean or Median Hepcidin Levels in Controls	Main Study Results
Kanda et al., 2008 [31]	Asia (Japan)	31 (W: ND M: ND)	51	13 AML, 8 NHL, 3 MDS, 3 ATL, 2 ALL, 1 HL, 1 MM	Yes (5 autologous; 26 allogeneic)	LC/ESI-MS/MS	Blood (serum)	42.8 ng/mL (one week pre-HCT)	17 (W:1 M:16)	31	19.05 ng/mL	(1) hepcidin levels were higher in patients than in controls (2) hepcidin levels increased until the first week post-HCT, then decreased to the fourth week after HCT
								232.5 ng/mL (one week post-HCT)				
Eisfeld et al., 2011 [32]	Europe (Germany)	42 (W:19 M: 23)	57	AML	Yes (42 allogeneic)	ELISA (Intrinsic Hepcidin IDx™ ELISA Kit, Intrinsic Life Sciences, La Jolla, CA, USA)	Blood (serum)	358 ng/mL (ten days pre-HCT)	21 (W:15 M:6)	57	52.1 ng/mL	(1) hepcidin levels were higher in patients than in controls (2) pre- and post-HCT hepcidin levels were similar
								398 ng/mL (three months post-HCT				
Cheng et al., 2012 [20]	Asia (China)	32 (W:13 M:19)	Three groups of patients *: A—29 (*n =* 10) B—33 (*n =* 15) C—34 (*n =* 7)	AL ^†^	No	ELISA (Uscn Life Science Inc, Wuhan, China)	Blood (serum)	A—343.447 ng/mL B—523.758 ng/mL C—486.176 ng/mL (all before treatment)	11 (W:4 M:7)	35	141.098 ng/mL	(1) hepcidin levels were higher in patients than in controls, regardless of patients’ iron storage (2) hepcidin levels decreased during complete or partial remission
								685.633 ng/mL (before treatment; *n =* 20)				
								485.438 ng/mL (during complete or partial remission; *n =* 20)				
Chen et al., 2013 [34]	Asia (China)	57 (W: 26 M: 31)	49	27 HCT (due to hematologic tumors), 18 liver transplantation, 12 kidney transplantation	Yes (27 autologous or allogeneic)	ELISA (DRG Instruments GmbH, Marburg, Germany)	Blood (serum)	38.31 ng/mL (pretransplant)	50 (W:ND M: ND)	ND	18.70 ng/mL	(1) hepcidin levels were higher in patients than in controls (2) hepcidin levels increased until the first week after transplantation, then decreased to the fourth week after transplantation (3) pretransplant hepcidin as a biomarker of invasive fungal disease
								51.82 ng/mL (one week before transplantation in the high-hepcidin group; *n =* 19) ^‡^				
								129.60 ng/mL (one week after transplantation in the high-hepcidin group; *n =* 19)				
Ragab et al., 2016 [36]	Africa (Egypt)	40 (W: 13 M: 27)	Two groups of patients ^#^: I—5 (*n =* 20) II—5 (*n =* 20)	ALL	No	ELISA (EIAab^®^ Human Hepcidin ELISA kit, EIAab Science INC, Wuhan, China)	Blood (serum)	Group I: 387.6 ng/mL (at diagnosis) 221.5 ng/mL (after remission)	20 (W:6 M:14)	6	69.8 ng/mL	(1) hepcidin levels were higher in patients than in controls, regardless of the stage of the disease (2) hepcidin levels decreased during remission (3) hepcidin levels were the lowest during maintenance therapy
								Group II: 181.9 ng/mL (during maintenance therapy)				
Yavuz et al., 2017 [37]	Asia (Turkey)	58 (W: 26 M: 32)	10	28 sarcomas, 11 ALL, 10 solid tumors, 9 lymphomas ^$^	No	ELISA (DRG Instruments GmbH, Marburg, Germany)	Blood (serum)	Sarcomas 34.51 ng/mL (at diagnosis) 17.92 ng/mL (at remission)	17 (W: 8 M:9)	9	6.98 ng/mL	(1) hepcidin levels were higher in patients than in controls, regardless of the stage of the disease
								Lymphomas 24.83 ng/mL (at diagnosis) 21.13 ng/mL (at remission)				
								ALL 58.45 ng/mL (at diagnosis) 50.81 ng/mL (at remission)				
								Solid tumors 43.82 ng/mL (at diagnosis) 35.60 ng/mL (at remission)				
Łęcka et al., 2021 [21]	Europe (Poland)	67 (W: 30 M: 37)	7	21 AL de novo, 25 AL after intensive treatment, 21 HCT ^€^	Yes (3 autologous; 18 allogeneic)	ELISA (Intrinsic Hepcidin IDx™ ELISA Kit, Intrinsic Life Sciences, La Jolla, CA, USA)	Blood (serum)	158.50 ng/mL (AL de novo)	18 (W:10 M: 8)	8	30.61 ng/mL	(1) hepcidin levels were higher in patients than in controls, regardless of the stage of the disease (2) hepcidin levels were the highest post-HCT
								106.60 ng/mL (AL after intensive therapy)				
								278.30 ng/mL (one month post-HCT)				

* Patients were divided into three groups (A, B, C) according to the degree of the extracellular iron (EI) store and the value of the intracellular iron (II) store. ^†^ FAB criteria: 2 M1, 4 M2, 5 M3, 6 M4, 3 M5, 12 ALL. ^‡^ High-hepcidin group = patients with serum hepcidin levels greater than 40 ng/mL. We wrote ‘one week before transplantation’ on purpose, not ‘one week before HCT’ because the patient group included those who underwent HCT and those who underwent liver and kidney transplants. ^#^ Patients were divided into two groups (I, II): group I (newly diagnosed ALL; *n* = 20), group II (patients with ALL in the maintenance phase of therapy; *n =* 20). ^$^ Sarcoma (11 Ewing’s sarcoma, 11 osteosarcoma, 6 rhabdomyosarcoma); solid tumors (4 neuroblastoma, 3 central nervous system tumors, 2 Wilms tumor, 1 hepatoblastoma); lymphoma (5 NHL, 4 HL). ^€^ HCT group: 8 acute myeloid leukemia, 6 acute lymphoblastic leukemia, 2 neuroblastoma, 1 myelodysplastic syndrome, 1 severe aplastic anemia, 1 severe congenital neutropenia, 1 anaplastic large B-cell lymphoma, 1 Ewing sarcoma. Abbreviations: AL = acute leukemia; ALL = acute lymphoblastic leukemia; AML = acute myeloid leukemia; ATL = adult T-cell leukemia; ELISA = enzyme-linked immunosorbent assay; HL = Hodgkin lymphoma; HCT = hematopoietic cell transplantation; LC/ESI-MS/MS = liquid chromatography-electrospray ionization tandem mass spectrometry; M = man; MDS = myelodysplastic syndrome; MM = multiple myeloma; ND = not determined; NHL = non-Hodgkin lymphoma; W = woman.

**Table 2 cancers-14-04936-t002:** Characteristics of the four cohort studies and the two cross-sectional studies included in a systematic review.

First Author, Year	Region of Origin	Number of Patients	Mean or Median Age of Patients [Years]	Clinical Diagnosis	HCT	Hepcidin Assay	Type of Biological Material	Mean or Median Hepcidin Levels in Patients	Follow-Up	Main Study Results
Kanda et al., 2009 [22]	Asia (Japan)	55 (W: 28 M: 27)	47 (whole cohort) Two groups of patients *: Low-hepcidin—47.5 (*n =* 38) High-hepcidn—47 (*n =* 17)	23 AML, 9 MDS/ MPN, 9 NHL, 8 ALL, 5 ATL, 1 HL	Yes (47 allogeneic)	LC/ESI-MS/MS	Blood (serum)	21.6 ng/mL (pre-HCT)	100 days post-HCT	(1) high hepcidin levels (≥ 50 ng/mL) are associated with an increased risk of bacterial infection post-HCT (2) hepcidin as a biomarker of bacterial infection post-HCT
Armand et al., 2011 [33]	North America (USA)	48 (ND)	47	29 AML, 11 ALL, 8 MDS	Yes (48 allogeneic)	MALDI-TOF MS	Blood (plasma or serum) ^†^ Urine	59 ng/mL (blood, pre-HCT, *n =* 39)	-	(1) blood hepcidin levels are correlated to other iron parameters
								110 ng/mg creatinine (urine, *n =* 33)		
Naoum et al., 2016 [35]	South America (Brazil)	25 (W: 15 M: 10)	46	13 MM, 8 ML, 3 AL, 1 seminoma	Yes (25 autologous)	ELISA (DRG Instruments GmbH, Marburg, Germany)	Blood (serum)	25.1 ng/mL (before the start of conditioning)	engraftment ^#^	(1) hepcidin levels were higher before SC infusion and on engraftment than before the start of conditioning
								40.0 ng/mL (before SC infusion)		
								39.1 ng/mL (on engraftment)		
Sakamoto et al., 2017 [24]	Asia (Japan)	166 (W: 74 M: 92)	49.5 (whole cohort) Two groups of patients ^‡^: Low-hepcidin—47 (*n =* 83) High-hepcidn—51 (*n* = 83) ^‡^	103 MMa, 63 LM	Yes (166 allogeneic)	SELDI-TOF MS	Blood (serum)	7.8 ng/mL ^$^	46.8 months (median)	(1) high hepcidin levels (≥35 ng/mL) are associated with lower overall survival post-HCT (2) high hepcidin levels are associated with a lower incidence of platelet engraftment post-HCT
								35.0 ng/mL (pre-HCT)		
Wermke et al., 2018 [38]	Europe (Germany)	112 (W: 47 M:65)	62 (whole cohort) Two groups of patients ^€^: eLPI μmol/L ≤ 0.4—62 (*n =* 85) eLPI μmol/L > 0.4 —62 (*n* = 27)	90 AML, 22 MDS	Yes (112 allogeneic)	ELISA (DRG Instruments GmbH, Marburg, Germany)	Blood (serum)	77 ng/mL (whole cohort; pre-HCT)	373 days (median)	(1) hepcidin levels were higher in the eLPI > 0.4 μmol/L group pre-HCT and on day 21 post-HCT
								eLPI ≤ 0.4 μmol/L: 70 ng/mL (pre-HCT) 64 ng/mL (on the day of HCT) 81 ng/mL (on day 21 post-HCT)		
								eLPI > 0.4 μmol/L: 103 ng/mL (pre-HCT) 83 ng/mL (on the day of HCT) 127 ng/mL (on day 21 post-HCT)		
Wande et al., 2020 [39]	Asia (Indonesia)	48 (W: 17 M: 31)	Three groups of patients: induction phase—6.8 (*n =* 16) consolidation phase—9.7 (*n =* 16) maintenance phase—7.8 *(n =* 16)	48 ALL	No	ELISA (Bioassay Technology Laboratory, Jiaxing, China)	Blood (serum)	7.545 ng/mL (induction phase)	-	(1) hepcidin levels vary depending on disease state
								1.728 ng/mL (consolidation phase)		
								0.210 ng/mL (maintenance phase)		

* Patients were divided into two groups: low-hepcidin group (hepcidin levels lower than 50 ng/mL; *n =* 38), high-hepcidin group (hepcidin levels greater than 50 ng/mL; *n =* 17). ^†^ In the materials and methods section, the authors reported that they measured plasma levels of hepcidin, but the results report the levels of hepcidin in serum samples. ^‡^ Patients were divided into two groups: low-hepcidin group (hepcidin levels lower than 35 ng/mL; *n =* 83), high-hepcidin group (hepcidin levels greater than 35 ng/mL; *n =* 83). ^#^ Engraftment was defined as the first of 3 consecutive days with an absolute neutrophil count of at least 0.5 × 10^9^/L. ^$^ In the results section, the authors presented the hepcidin levels in the healthy volunteers; however, this group was not described in materials and methods, and there are no details regarding this analysis; hence, we arbitrarily concluded that the study is a cohort. ^€^ Patients were divided into two groups: eLPI (enhanced labile plasma iron) ≤ 0.4 μmol/L (*n =* 85), eLPI > 0.4 μmol/L (*n =* 27). Abbreviations: AL = acute leukemia; ALL = acute lymphoblastic leukemia; AML = acute myeloid leukemia; ATL = adult T-cell leukemia; ELISA = enzyme-linked immunosorbent assay; HL = Hodgkin lymphoma; HCT = hematopoietic cell transplantation; LC/ESI-MS/MS = liquid chromatography-electrospray ionization tandem mass spectrometry; LM = lymphoid malignancies; M = man; MALDI-TOF MS = matrix assisted laser desorption/ionization time-of-flight mass spectrometry; MDS = myelodysplastic syndrome; ML = malignant lymphoma; MM = multiple myeloma; MMa = myeloid malignancies; ND = not determined; NHL = non-Hodgkin lymphoma; SELDI-TOF MS = surface enhanced laser desorption/ionization time-of-flight mass spectrometry; SC = stem cell; W = woman.

**Table 3 cancers-14-04936-t003:** The blood hepcidin levels reported in patients with AL/HCT on the background of serum ferritin levels and packed red blood cells (PRBCs).

First Author, Year	Mean or Median Hepcidin Levels in Patients *	Mean or Median Ferritin Levels in Patients ^†^ Reference Range: 1.2–20.0 μg/dL	Mean or Median Units of PRBCs
Kanda et al., 2008 [31]	42.8 ng/mL (one week pre-HCT)	726.3 μg/dL (one week pre-HCT)	ND
	232.5 ng/mL (one week post-HCT)		
Kanda et al., 2009 [22]	21.6 ng/mL (pre-HCT)	664 μg/dL (low hepcidin group)	ND
		1551 μg/dL (high hepcidin group)	
Eisfeld et al., 2011 [32]	358 ng/mL (ten days pre-HCT)	194.5 μg/dL (pre-HCT)	22 (pre-HCT)
	398 ng/mL (three months post-HCT)	226 μg/dL (post-HCT)	30 (post-HCT)
Armand et al., 2011 [33]	59 ng/mL (pre-HCT)	154.9 μg/dL	20 (pre-HCT)
Cheng et al., 2012 [20]	A—343.447 ng/mL B—523.758 ng/mL C—486.176 ng/mL (all before treatment)	A—62. 806 μg/dL B—94.964 μg/dL C—77.381 μg/dL (all before treatment)	ND
	685.633 ng/mL (before treatment)	105.082 μg/dL (before treatment)	
	485.438 ng/mL (during complete or partial remission)	61.437 μg/dL (during complete or partial remission)	
Chen et al., 2013 [34]	38.31 ng/mL (pretransplant) ^‡^	The authors did not present numerical values for ferritin but observed its high serum levels.	ND
	51.82 ng/mL (one week before transplantation in the high-hepcidin group)		
	129.60 ng/mL (one week after transplantation in the high-hepcidin group)		
Naoum et al., 2016 [35]	25.1 ng/mL (before the start of conditioning)	73.3 μg/dL (before the start of conditioning)	3 (from SC infusion to engraftment)
	40.0 ng/mL (before SC infusion)	78.2 μg/dL (before SC infusion)	
	39.1 ng/mL (on engraftment)	77.8 μg/dL (on engraftment)	
Sakamoto et al., 2017 [24]	35.0 ng/mL (pre-HCT)	69.4 μg/dL (pre-HCT)	
Ragab et al., 2016 [36]	Group I: 387.6 ng/mL (at diagnosis) 221.5 ng/mL (after remission)	Group I: 126.5 μg/dL (at diagnosis) 79.3 μg/dL (after remission)	ND
	Group II: 181.9 ng/mL (during maintenance therapy)	Group II: 60.4 μg/dL (during maintenance therapy)	
Yavuz et al., 2017 [37]	Sarcomas 34.51 ng/mL (at diagnosis) 17.92 ng/mL (at remission)	ND	ND
	Lymphomas 24.83 ng/mL (at diagnosis) 21.13 ng/mL (at remission)		
	ALL 58.45 ng/mL (at diagnosis) 50.81 ng/mL (at remission)		
	Solid tumors 43.82 ng/mL (at diagnosis) 35.60 ng/mL (at remission)		
Wermke et al., 2018 [38]	77 ng/mL (whole cohort; pre-HCT)	1731 μmol/L ^#^ (whole cohort)	18 (whole cohort)
	eLPI ≤ 0.4 μmol/L: 70 ng/mL (pre-HCT) 64 ng/mL (on the day of HCT) 81 ng/mL (on day 21 post-HCT)	eLPI ≤ 0.4 μmol/L: 1563 μmol/L	eLPI ≤ 0.4 μmol/L: 18
	eLPI > 0.4 μmol/L: 103 ng/mL (pre-HCT) 83 ng/mL (on the day of HCT) 127 ng/mL (on day 21 post-HCT)	eLPI > 0.4 μmol/L: 2425 μmol/L	eLPI > 0.4 μmol/L: 22
Wande et al., 2020 [39]	7.545 ng/mL (induction phase)	135.307 μg/dL (induction phase)	ND
	1.728 ng/mL (consolidation phase)	175.826 μg/dL (consolidation phase)	
	0.210 ng/mL (maintenance phase)	74.977 μg/dL (maintenance phase)	
Łęcka et al., 2021 [21]	158.50 ng/mL (AL de novo)	23.85 μg/dL (AL de novo)	1
	106.60 ng/mL (AL after intensive therapy)	73.9 μg/dL (AL after intensive therapy)	9
	278.30 ng/mL (one month post-HCT)	367.0 μg/dL (one month post-HCT)	23

* The results are partially repeated with those presented in Table 1 and Table 2; however, the re-presentation of the hepcidin levels facilitates its comparison with those of ferritin. ^†^ The units of the ferritin levels are standardized and presented as micrograms per deciliter (μg/dL). The reference values were based on data from the “*WHO guideline on use of ferritin concentrations to assess iron status in individuals and populations*” (https://www.who.int/publications/i/item/9789240000124 (accessed on 4 July 2022)). ^‡^ We wrote ‘pre-transplant’ on purpose, not pre-HCT because the patient group included those who underwent HCT and those who underwent liver and kidney transplants. ^#^ The unit in which the authors reported the ferritin levels was left. However, it seems that a mistake was made here (e.g., when calculating the levels for all patients (*n =* 112), we receive extremely high values, i.e., 1730.0 μmol/L = 76985000.0 μg/dL). Abbreviations: AL = acute leukemia; ALL = acute lymphoblastic leukemia; eLPI = enhanced labile plasma iron; HCT = hematopoietic cell transplantation; ND = not determined; PRBC = packed red blood cells; SC = stem cell.

## Data Availability

The data can be shared up on request.

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
