# Peer review of "Hepcidin in Children and Adults with Acute Leukemia or Undergoing Hematopoietic Cell Transplantation: A Systematic Review"

_cancers, 2022, doi:10.3390/cancers14194936_

Round 1

Reviewer 1 Report

   In this review, the authors analyzed hepcidin levels in patients with acute leukemia or who underwent hematopoietic cell transplantation thorough a systematic review of observational studies.  After all, they obtained 4 findings in terms of the relationship of hepcidin levels and disease status,  age, or ferritin levels.  Although they carefully selected the eligible studies, too few studies remained and also too much variations are seen among the selected studies.  At the present form, the reader cannot obtain any valuable information from this review, since above 4 significant findings are already suggested in the original reports. 

Author Response

REVIEWER #1

In this review, the authors analyzed hepcidin levels in patients with acute leukemia or who underwent hematopoietic cell transplantation thorough a systematic review of observational studies.  After all, they obtained 4 findings in terms of the relationship of hepcidin levels and disease status,  age, or ferritin levels.  Although they carefully selected the eligible studies, too few studies remained and also too much variations are seen among the selected studies.  At the present form, the reader cannot obtain any valuable information from this review, since above 4 significant findings are already suggested in the original reports.

REPLY TO REVIEWER #1 COMMENTS

Many thanks for the reviewer’s comments on our manuscript. The research topic we undertook is extraordinarily topical and extremely important from a clinical point of view due to the problem of iron overload in patients with acute leukemia (AL) or undergoing hematopoietic cell transplantation (HCT). For this reason, we undertook a challenging task, in our opinion, namely a systematic review of what is known about the role of hepcidin, the central controller of iron metabolism, in the patients mentioned above. We adopted the regime methodology of our literature review, which was clearly defined in the materials and methods. The methodology we adopted was approved by the scientific team of PROSPERO, which is an international database of prospectively registered systematic reviews in medicine, run by the Centre for Reviews and Dissemination, University of York, UK. A similar methodology has been used in our other systematic reviews, for example, published in Thrombosis and Haemostasis, a journal with significant IF (Słomka A. et al. Thromb Haemost. 2020;120(5):815-822;  Słomka A. et al. Thromb Haemost. 2021;121(9):1181-1192). In addition, in line with PRISMA principles, all manuscripts that did not meet our inclusion criteria were described in detail in the supplementary materials, explaining why they were not included in the systematic review. Our study provides a comprehensive overview of the role of hepcidin in AL and HCT patients. The reader can effectively analyze observational studies in this field without the need to painstakingly investigate many publications. The tables also make it easier to understand this topic. We agree with the reviewer that there is noteworthy variability between patients in the included manuscripts. For this reason, we did not carry out a meta-analysis of the obtained results, what is stated in the results of our systematic review. Despite this, the reader can efficiently analyze what relationships have been shown so far between the levels of blood hepcidin and the course or treatment of AL and HCT patients. Our study also clearly shows the need to harmonize the methods of determining the levels of hepcidin in clinical conditions, so that this parameter becomes a useful biomarker in the prognosis in AL or in prognosis responding to HCs transplantation. Additionally, conclusions from our systematic review bring strong information for the readers, as were obtained from analysis of a large number of papers. We believe these explanations are sufficient and do not require any changes in the manuscript.

Reviewer 2 Report

The text by Slomka A et al. reviews current literature on Hepcidin in patients with acute leukemia or stem cell transplantation.

The authors describe in detail 7 case-control studies, and subsequently 4 cohort and 2 cross sectional studies.

The study methodology, analysis and conclusions are adequate with respect to the study results. The bibliography is up to date.

Problems

1) since the population included in the study is so heterogeneous, it is difficult to obtain other conclusions than those described by the authors.

2) unfortunately the data of the review do not provide the volumes of blood transfused and this certainly represents a limitation of the study (in this case the corresponding authors had to be contacted)

3) the authors report that the phase of the disease in both pediatric and adult patients affects the serum level of hepcidin, unfortunately the data reported are not detailed (in this case the corresponfing authors had to be contacted)

Despite the great effort made by the authors, this review does not add much to the present literature.

Author Response

REVIEWER #2

The text by Slomka A et al. reviews current literature on Hepcidin in patients with acute leukemia or stem cell transplantation.

The authors describe in detail 7 case-control studies, and subsequently 4 cohort and 2 cross sectional studies.

The study methodology, analysis and conclusions are adequate with respect to the study results. The bibliography is up to date.

Problems

1) since the population included in the study is so heterogeneous, it is difficult to obtain other conclusions than those described by the authors.

2) unfortunately the data of the review do not provide the volumes of blood transfused and this certainly represents a limitation of the study (in this case the corresponding authors had to be contacted)

3) the authors report that the phase of the disease in both pediatric and adult patients affects the serum level of hepcidin, unfortunately the data reported are not detailed (in this case the correspondfing authors had to be contacted)

Despite the great effort made by the authors, this review does not add much to the present literature.

REPLY TO REVIEWER #2 COMMENTS

Many thanks for the reviewer’s comments on our manuscript.

  1. The regime manuscript selection methodology we adopted allowed us to select those observational studies that strictly assess the role of hepcidin in patients with acute leukemia (AL) or undergoing hematopoietic cell transplantation (HCT). Our results have been carefully and scrupulously analyzed to make it facilitate for the reader to understand this complex topic. We agree with the reviewer that there is noteworthy variability between patients in the included manuscripts. For this reason, we did not carry out a meta-analysis of the obtained results, what is stated in the results of our systematic review. Despite this, the reader can effectively analyze what relationships have been shown so far between the levels of blood hepcidin and the course or treatment of AL and HCT patients. By analyzing the tables, the reader can understand not only the cardinal role of hepcidin in the course of AL, but also prepare to undertake their research on this protein in the course of the diseases mentioned above or on the impact of treatment methods on hepcidin levels and its role in predicting patients’ outcomes. Our study also clearly shows the need to harmonize the methods of determining the levels of hepcidin in clinical conditions, so that this parameter becomes a useful biomarker in the prognosis of the course of AL or in prognosis responding to HCs transplantation.

2 and 3. We thank the reviewer for drawing our attention to these issues. Table 3 presents data on blood transfusions performed on patients in the included studies (mean or median units of PRBCs). In the case of the relationship between the disease phase and hepcidin levels, these data are presented in detail in Table 1 (assessment of blood hepcidin levels depending on the stage of AL treatment and evaluation of levels before or after HCT).

Regarding contacting the authors of selected manuscripts, this procedure is not a universal feature of systematic reviews published in top journals and the Cochrane Library (Mullan R.J. et al. J Clin Epidemiol. 2009;62(2):138-142). We also did not anticipate such a stage in registering our review in PROSPERO. Moreover, some authors suggest the low usefulness of such a procedure in systematic reviews (Selph SS et al. Syst Rev. 2014;3:107) and its instead help in quantitative analysis (Sho T et al. Journal of Psychology. 2020; 228(1):50-61), which we did not perform due to the high heterogeneity in the included studies.

Additionally, conclusions from our systematic review bring strong information for the readers, as were obtained from analysis of a large number of papers. We believe these explanations are sufficient and do not require any changes in the manuscript.

Reviewer 3 Report

In this review the authors summarize the knowledge concerning the importance of hepcidine levels in the setting of acute leukemia and stem cell transplantation. Although for some time it has been known that pre-transplantation hyperferritinemia (mostly due to iron overload) has a negative prognostic role and is associated with not only decreased overall survival and progression-free survival,but higher incidence of non-relapse mortality and bloodstream infections, as well. Hepcidine, a master regulator of iron homeostasis may potentially be an even better indicator of these parameters. After a thorough literature search, 13 relevant publications were dissected and analyzed in detail. It was found that hepcide levels - not entirely unexpectedly for an acute phase protein - correlate not only with iron overload but are also influenced by various other parameters, such as type of antineoplastic therapy and transplantation type and stage.

Unfortunately, althogh the authors did try to identify, the medical relavance of the observations remain unclear, further investigation is warranted,

Author Response

REVIEWER #3

In this review the authors summarize the knowledge concerning the importance of hepcidine levels in the setting of acute leukemia and stem cell transplantation. Although for some time it has been known that pre-transplantation hyperferritinemia (mostly due to iron overload) has a negative prognostic role and is associated with not only decreased overall survival and progression-free survival,but higher incidence of non-relapse mortality and bloodstream infections, as well. Hepcidine, a master regulator of iron homeostasis may potentially be an even better indicator of these parameters. After a thorough literature search, 13 relevant publications were dissected and analyzed in detail. It was found that hepcide levels - not entirely unexpectedly for an acute phase protein - correlate not only with iron overload but are also influenced by various other parameters, such as type of antineoplastic therapy and transplantation type and stage.

Unfortunately, although the authors did try to identify, the medical relavance of the observations remain unclear, further investigation is warranted,

REPLY TO REVIEWER #3 COMMENTS

Many thanks for the reviewer’s comments on our manuscript. The research topic we undertook is extraordinarily topical and extremely important from a clinical point of view due to the problem of iron overload in patients with acute leukemia (AL) or undergoing hematopoietic cell transplantation (HCT). For this reason, we undertook a challenging task, in our opinion, namely a systematic review of what is known about the role of hepcidin, the central controller of iron metabolism, in the patients mentioned above. We adopted the regime methodology of our literature review, which was clearly defined in the materials and methods. The methodology we adopted was approved by the scientific team of PROSPERO, which is an international database of prospectively registered systematic reviews in medicine, run by the Centre for Reviews and Dissemination, University of York, UK. A similar methodology has been used in our other systematic reviews, for example, published in Thrombosis and Haemostasis, a journal with significant IF (Słomka A. et al. Thromb Haemost. 2020;120(5):815-822;  Słomka A. et al. Thromb Haemost. 2021;121(9):1181-1192). Our study provides a comprehensive overview of the role of hepcidin in AL and HCT patients. The reader can effectively analyze observational studies in this field without the need to painstakingly investigate many publications. The tables also make it easier to understand this topic. Despite noteworthy variability between patients in the included manuscripts, the reader can efficiently analyze what relationships have been shown so far between the levels of blood hepcidin and the course or treatment of AL and HCT patients. Our study also clearly shows the need to harmonize the methods of determining the levels of hepcidin in clinical conditions, so that this parameter becomes a useful biomarker in the prognosis in AL or in prognosis responding to HCs transplantation. Additionally, conclusions from our systematic review bring strong information for the readers, as were obtained from analysis of a large number of papers. We believe these explanations are sufficient and do not require any changes in the manuscript.

Round 2

Reviewer 1 Report

   The hepcidin levels correlate to the ferritin levels and iron overload status. In daily clinical practice, it is easier to measure the ferritin levels and check the iron overload status. So, in practical meaning, clinicians will not select the measurement of hepcidin levels.

Reviewer 2 Report

Authors full replied to the criticism. Please add these point into the text

Reviewer 3 Report

I accept your reasons explained in the rebuttal.